# Longitudinal Associations between Teacher-Student Relationships and Prosocial Behavior in Adolescence: The Mediating Role of Basic Need Satisfaction

**DOI:** 10.3390/ijerph192214840

**Published:** 2022-11-11

**Authors:** Guoqiang Wu, Lijin Zhang

**Affiliations:** 1School of Psychology, Shaanxi Normal University, Xi’an 710062, China; 2School of Teachers Education, Xi’an University, Xi’an 710065, China; 3Shaanxi Provincial Key Research Center of Child Mental and Behavioral Health, Xi’an 710062, China; 4Shaanxi Key Laboratory of Behavior and Cognitive Neuroscience, Xi’an 710062, China

**Keywords:** teacher-student relationships, prosocial behavior, basic psychological need satisfaction, relationships motivational theory

## Abstract

The quality of teacher-student relationships has been shown to relate to adolescents’ prosocial behavior, but the motivational mechanisms underlying this association remained unclear. Based on relationships motivation theory (RMT), we examined whether the associations between teacher-student relationships (closeness and conflict) and prosocial behavior are bidirectional, and the mediating role of basic psychological need satisfaction (autonomy, competence, and relatedness need satisfaction) in these links. Data were collected from a sample of 438 secondary school students who completed a survey at two-time points over eight months. The cross-lagged autoregressive model revealed that the relation between close teacher-student relationship and prosocial behavior was bidirectional over time. Moreover, relatedness need satisfaction mediated the positive effect of close teacher-student relationship and the negative effect of teacher-student relationship conflict on adolescents’ prosocial behavior. This study highlights the importance of close teacher-student relationship and relatedness need satisfaction to foster adolescents’ prosocial behavior.

## 1. Introduction

Prosocial behavior refers to social behavior that aims to meet social expectations and benefit others and society when interacting with others [1]. As an essential mission for adolescents’ socialization development, prosocial behavior has been found to be associated with adolescents’ various positive outcomes, such as better peer relationships [2], academic achievement [3], and social adaptation [4]. Teachers play a critical role in promoting adolescents’ development of social behavior through their interaction with students [5]. Previous research has shown that the quality of students’ relationships with their teachers has been identified as an essential factor influencing adolescents’ prosocial behavior [6,7]. However, to date, most of the research on teacher-student relationships has focused on the unidirectional effect on students’ prosocial behavior in childhood and early adolescence and has primarily relied on cross-sectional designs. Few studies have examined the longitudinal associations between positive and negative aspects of teacher-student relationships and adolescents’ prosocial behavior in secondary education, and the motivational mechanisms underlying these associations. To address these issues, we first examined the reciprocal effects between teacher-student relationships (closeness and conflict) and adolescents’ prosocial behavior over time. Second, based on the relationships motivation theory (RMT) [8], we explored the underlying motivational mechanisms of basic psychological need satisfaction (autonomy, competence, and relatedness need satisfaction) in these relationships.

### 1.1. Effects of Teacher-Student Relationships on Prosocial Behavior

Research on the quality of teacher-student relationship has received extensive attention. Wubbels and Brekelmans [9] introduced the model of interpersonal diagnosis of personality into teaching context and defined the teacher-student relationship as two main dimensions: Influence (reflecting the extent to which teacher directs and controls the communication) and Proximity (reflecting the degree of cooperation between the teacher and the student). Moreover, in the pattern level, research further classified these dimensions into eight types of patterns of interpersonal relationships with student: authoritative, tolerant/authoritative, tolerant, directive, uncertain/aggressive, uncertain/tolerant, drudging and repressive. However, most of the studies adopted attachment theory and defined the teacher-student relationships as two distinct dimensions: closeness and conflict. Closeness refers to the student’s perceptions of warmth, feeling cared for and respected by their teachers, whereas conflict typically involves discordant interaction, feelings of distrust, negativity in the relationship and insecure attachment [10].

The teacher-student relationship quality has significant implications for adolescents’ prosocial behavior [11]. According to attachment theory [12], a supportive and close teacher-student relationship enables students to form a positive internalized working model toward the self and others, allowing them to feel more security, competence, and a sense of trust for others. Furthermore, these experiences, in turn, may foster adolescents to provide support and care for others. In addition, adolescents who develop a positive relationship with teachers tend to perceive their teacher as a secure base that can provide them emotional support, convey positive expectations, and reinforce positive behavior in school [13]. Therefore, adolescents are likely to engage in more prosocial behavior in order to develop and sustain positive relationship with their teacher. Further, recent studies have supported this view by showing that close relationship with teachers positively predict lower discipline problems, fewer aggressive behavior, and more prosocial behavior 4 years later [11].

In contrast, conflict in student-teacher relationship has been consistently related to a range of maladaptive outcomes such as increased aggressive behavior [14] and higher levels of peer victimization [15]. These results suggest that having a conflictual relationship with teachers allows students to experience feelings of insecurity, more negative emotions such as distress [16], and lower levels of social competence [17,18], which may reduce their intention to display prosocial behavior. Additionally, students in conflictual relationships with their teachers cannot gain support from their teachers in school and are not likely to engage in prosocial behavior. In light of these views, empirical research has demonstrated that conflictual teacher-student relationship is negatively related to students’ prosocial behavior in elementary [6,19,20] and middle school [18]. These findings may suggest a promoting effect of the close teacher-student relationship and an inhibiting effect of the conflictual teacher-student relationship on prosocial behavior. However, the longitudinal evidence for this link among adolescents is scarce.

### 1.2. Effects of Prosocial Behavior on Teacher-Student Relationship

According to socialization theory, students’ characteristics and behaviors can unconsciously influence how teachers interact with their students [10,21]. In this sense, teachers are likely to respond positively to their students in a supportive way when students display more positive social behavior at school, such as prosocial behavior. Teachers tend to form conflictual and unsupportive relationships with students who demonstrate more negative social behaviors such as aggressive behavior. Based on this perspective, research has shown a positive association of prosocial behavior with close teacher-student relationship in children and mid-adolescence [22,23]. These findings are aligned with the student-driven model in which student characteristics can influence the teacher-student relationship quality [24]. However, the empirical evidence for the predictive role of students’ prosocial behavior on conflictual relationships with teachers is scarce and mixed. Some reported a negative association of prosocial behavior with teacher-student relationship conflict in childhood [25], but some did not [20]. These inconsistent findings may be attributed to the difference in educational level, as previous research has mostly focused on childhood or early adolescence and has not explored the longitudinal effect of students’ prosocial behavior on the teacher-student relationships in middle school.

Although previous research has documented the unidirectional causal relationship between teacher-student relationships and prosocial behavior, little research has investigated their reciprocal effects in secondary school students. The transactional model holds that an individual’s social development arises from the reciprocal interaction between individuals and their environment [20]. Specifically, school social context can affect students’ social characteristics and behavior, which can actively react to their environment. Based on this view, the potential reciprocal relations between teacher-student relationships and prosocial behavior over time may exist. However, the result regarding this reciprocal link is scarce and inconclusive. For example, a short-term longitudinal study found a reciprocal relationship between closeness and conflict with preschool children’s prosocial behavior [25], while others did not find it in early adolescence [26].

### 1.3. The Mediating Role of Psychological Need Satisfaction

#### 1.3.1. Teacher-Student Relationship and Psychological Need Satisfaction

We applied the relationships motivation theory (RMT), the sixth mini-theory of self-determination theory (SDT), to examine the basic psychological need satisfaction (relatedness, autonomy, and competence) as underlying motivational mechanism in the links between teacher-student relationships and adolescents’ prosocial behavior.

The central tenet of SDT is that people have a natural tendency to satisfy the basic psychological needs for relatedness (i.e., feelings of care for and being cared for by others, and establishing caring and emotion supportive social relationships with significant others), autonomy (i.e., the experience of a sense of control for one’s own behavior and a sense of freedom), and competence (i.e., needs for feeling effective and confident in carrying out an activity in hand), all of which has been defined as motivational construct and have motivational effects on individuals’ social development and well-being [27,28]. RMT proposes that satisfaction of the need for relatedness is intrinsic and essential to developing and maintaining a high-quality close interpersonal relationship with significant others (i.e., teacher, parent, coach) in one’s life. Furthermore, when all three basic psychological needs are satisfied simultaneously within the relationship, people can sustain high-quality relational bonds with others [8]. Hence, the satisfaction of adolescents’ psychological needs for autonomy, competence, and relatedness contributes to a high-quality relationship with their teachers.

Moreover, high-quality intimate relationships can, in turn, satisfy adolescents’ basic psychological needs. According to SDT, when a positive teacher-student relationship can satisfy one’s psychological needs, the other two psychological needs can be satisfied simultaneously. In contrast, low-quality close relationships with teachers can hinder and undermine adolescents’ satisfaction of basic psychological needs [8,29]. Thus, RMT suggests a reciprocal relationship between the quality of close interpersonal relationships and basic psychological need satisfaction [8,30]. However, few studies have examined this perspective. Only one study to date has examined and found that a positive teacher-student relationship positively predicted subsequent adolescents’ autonomy, relatedness, and competence need satisfaction [31]. However, there is a lack of evidence about the role of the negative teacher-student relationships in adolescents’ basic need satisfaction.

#### 1.3.2. Psychological Need Satisfaction and Prosocial Behavior

We proposed a bidirectional relationship between basic need satisfaction and adolescents’ prosocial behavior. According to SDT, basic psychological needs are essential nutrients for well-being and social functioning, with need satisfaction relating to better social functioning and a lack of need satisfaction resulting in maladaptive social functioning [28]. Consistent with this claim, previous research has shown that need satisfaction is positively associated with students’ prosocial behavior [32,33]. Meanwhile, extant research has also provided initial evidence for the promoting effect of prosocial behavior on students’ need satisfaction. For instance, recent studies have demonstrated that students engaging in more prosocial acts reported high levels of relatedness satisfaction at school [22,34]. In addition, Miles and Upenieks [35] found that prosocial behavior is only positively associated with relatedness needs satisfaction and not significantly related to autonomy and competence needs. These findings have supported the bidirectional effect between need satisfaction and prosocial behavior. However, past research has mostly treated the satisfaction of three basic psychological needs as a global construct. Indeed, autonomy, competence and relatedness need satisfaction are distinct and not interchangeable [27,36]. As such, scholars have called for investigating the differential role of each basic need satisfaction in relation to various outcomes [37].

#### 1.3.3. Reciprocal Relationships between Teacher-Student Relationships, Need Satisfaction, and Prosocial Behavior

Based on the theory and literature above, we proposed that basic need satisfaction will mediate the relationship between teacher-student relationships and adolescents’ prosocial behavior. Specifically, a positive and close teacher-student relationship can facilitate the satisfaction of students’ basic psychological needs, further enhancing their prosocial behavior.

Although these propositions have not been examined by extant research, there is only indirect evidence for this claim. For instance, Jang et al. [38] found that early-semester perceived autonomy support indirectly influenced changes in adolescents’ prosocial behavior in late-semester via middle-semester need satisfaction, whereas teacher control thwarted the satisfaction of adolescents’ basic psychological needs, leading to low need satisfaction, sequentially decreasing students’ intention to engage in prosocial behavior. Autonomy support and teacher control are kind of teacher interpersonal behavior toward students, and are conceptually relevant with teacher-student relationships. However, previous research has primarily focused on the direct effect of the positive teacher-student relationship and need satisfaction on prosocial behavior. To our knowledge, no research has investigated the indirect effect of the negative teacher-student relationship on prosocial behavior through basic need satisfaction. The current study expands on these initial findings by testing whether need satisfaction mediates the link between close and conflict in teacher-student relationship and adolescents’ prosocial behavior.

### 1.4. The Present Study

Based on RMT, this study aimed to shed light on how positive and negative aspects of teacher-student relationships (closeness and conflict) related to adolescents’ prosocial behavior over time by investigating the mediating role of basic need satisfaction in this relationship. Although the reciprocal relationship between teacher-student relationships and prosocial behavior has been established, this link has not been examined in Chinese adolescent samples. Therefore, our first goal was to investigate whether the association between teacher-student relationships and prosocial behavior is reciprocal over time. Second, according to RMT and previous findings [38,39,40], we hypothesized that the positive effect of closeness on prosocial behavior and the negative effect of conflict on prosocial behavior would be partially mediated by basic need satisfaction (autonomy, competence, and relatedness). The hypotheses were proposed as follows:

**Hypothesis 1 (H1).** 
*Closeness at Time 1 relates positively to prosocial behavior at Time 2, and prosocial behavior at Time 1 relates positively to closeness at Time 2.*


**Hypothesis 2 (H2).** 
*Conflict at Time 1 relates negatively to prosocial behavior at Time 2, and prosocial behavior at Time 1 relates negatively to conflict at Time 2.*


**Hypothesis 3 (H3).** 
*Autonomy (H3a), competence (H3b), and relatedness (H3c) need satisfaction at Time 1 relate positively to prosocial behavior at Time 2.*


**Hypothesis 4 (H4).** 
*Autonomy (H4a), competence (H4b), and relatedness (H4c) need satisfaction mediate between closeness at Time 1 on prosocial behavior at Time 2.*


**Hypothesis 5 (H5).** 
*Autonomy (H5a), competence (H5b), and relatedness (H5c) need satisfaction mediated between conflict at Time 1 on prosocial behavior at Time 2.*


## 2. Method

### 2.1. Participants and Procedure

Participants were 438 secondary public-school students from 10 classrooms in three secondary schools in seventh and eighth grade (6 7th grade classrooms, 4 8th grade classrooms) in northwestern China. The schools were located in rural areas with an average socio-economic status. The total sample size was 480 students at Time 1 (266 7th, 214 8th graders) and 462 at Time 2 (239 7th, 211 8th graders). After excluding students with missing data, the final sample was 438 students (*n* = 257, boys, *n* = 181, girls) aged between 12 and 15 years (*M* = 13.53 years, *SD* = 0.87). Class size ranged from 44 to 55 students. We conducted a series of *t*-tests to test potential differences on the key variables between participants who dropped out (i.e., responded at T1 only) to those who were retained (i.e., completed at both T1 and T2). Results indicated no significant differences in terms of gender (χ^2^(1) = 0.65, *p* > 0.05), close (*t* (454) = −1.28, *p* > 0.05 and conflict (*t* (454) = 1.59, *p* > 0.05) teacher-student relationship, autonomy (*t* (454) = −0.73, *p* > 0.05), relatedness (*t* (454) = −1.68, *p* > 0.05) and competence (*t* (454) = −1.09, *p* > 0.05) need satisfaction, and prosocial behavior (*t* (454) = −1.66, *p* > 0.05). Therefore, these results suggested that this attrition might have not biased our results.

Data were collected at two points within the school year 2020–2021: Wave 1 (October 2020), at the start of the semester, and Wave 2 (June 2021), at the end of the semester, approximately 8 months apart. Consent to participate in the study were obtained from School’s principal, headmasters, teachers, and parents at each assessment point. Students were informed that they were voluntary and that information in the survey would be kept confidential, and this survey was not a test, and no right or wrong answers. Then, they were instructed to write down their student number and fill out a set of paper-and-pencil questionnaires during school time.

### 2.2. Measures

All variables in the current study are latent constructs. In order to reduce the complexity of the model and the number of estimated parameters in the model, we created several balanced parcels based on factor loadings by pairing higher with lower-loading items as indicators for each latent variable in the tested models [41].

#### 2.2.1. Teacher-Student Relationship (Self-Report)

The 13-item Chinese version of the Student-Teacher Relationship Scale was used to assess students’ perceptions of the quality of their relationship with the head teacher [42]. This scale is comprised of two subscales: closeness (7 items, e.g., “The relationship between my head teacher and I is close”; Cronbach’s α = 0.88 and 0.80 for T1 and T2, respectively) and conflict (6 items, e.g., “my head teacher does not care about me”; Cronbach’s α = 0.87 and 0.75 for T1 and T2, respectively). All items were scored on a 5-point Likert-type scale ranging from 1 (completely disagree) to 5 (completely agree). In this study, we created four parcels as indicators for closeness and three for conflict. The psychometrical properties (factor structure, reliability) of this scale have been supported by past research across countries [42].

#### 2.2.2. Psychological Need Satisfaction (Self-Report)

We used the Basic Need Satisfaction Scale (BNSS) to assess students’ perception of basic psychological need satisfaction for autonomy (6 items, e.g., “I am usually willing to express my thoughts and opinions”; Cronbach’s α = 0.81 and 0.80 for T1 and T2, respectively), competence (7 items, e.g., “I feel capable at what I do”; Cronbach’s α = 0.84 and 0.87 for T1 and T2, respectively), and relatedness(8 items, e.g., “I really like my classmate”; Cronbach’s α = 0.80 and 0.76 for T1 and T2, respectively). Respondents were asked to rate on a 5-point Likert-type scale ranging from 1 (totally disagree) to 5 (totally agree). Three parcels were created as indicators for autonomy need satisfaction, and four parcels for each latent construct of competence and relatedness need satisfaction. This scale has also been validated and demonstrated good psychometric property among Chinese adolescents [43].

#### 2.2.3. Prosocial Behavior (Peer-Report)

Adolescents’ prosocial behavior was assessed using peer nominations on six items (kind; willing to help others; really nice for classmates; cooperate with others; share with others; gives others the feeling that they belong to the group) [44,45]. Each student was asked to nominate up to five classmates who best fit each behavioral descriptor on a class roster. The score for prosocial behavior was calculated by summing the number of nominations on each item and standardized within the classroom. We created two parcels to serve as two indicators for latent constructs of prosocial behavior. This scale has shown adequate psychometric prosperities of validity and reliability in previous studies with Chinese adolescent students (e.g., Chen et al., 2000) [46]. In this study, Cronbach’s α were 0.80 and 0.85 for T1 and T2, respectively.

### 2.3. Statistical Analyses

In order to examine the longitudinal associations among teacher-student relationships, basic need satisfaction, and prosocial behavior, we employed the two-wave autoregressive cross-lagged model (ARC) to analyze the data. Cross-lagged structural equation modeling (SEM) enables control of the stability of the constructs while testing the causal relationship. All models in this study were tested with standardized coefficients using the maximum likelihood estimation method in Mplus [47].

A series of models were estimated to find out the best fit model: (1) the stability model (M0), which included the autoregressive paths between the same latent variables over time and the synchronous correlations between latent variables measured at the same time point (e.g., predictors, mediators, and outcome within T1), as well as the correlation between error terms of the same indicators over time; (2) the causality model (M1), in which the unidirectional paths from T1 predictors (close and conflictual teacher-student relationship) to T2 mediators (basic need satisfaction for autonomy, competence, and relatedness) and outcome (prosocial behavior), as well as from T1 mediators to T2 outcome are added to the stability model (M0) while controlling for stability effects and synchronous correlations of latent variables; (3) the reverse causation model (M2) represents the complete opposite of the causality model (M1): unidirectional paths from T1 mediators and T1 outcome to T2 predictors(close and conflictual teacher-student relationship) as well as T1 outcome (prosocial behavior) to T2 mediators; (4) a reciprocal model (M3), a combination of M1 and M2, represents the bidirectional paths between latent constructs.

We used four indices to evaluate model fit: the Comparative Fit Index (CFI), Tucker-Lewis Index (TLI), the Root Mean Square Error of Approximation (RMSEA), the confidence interval (CI), and Standardized Root Mean Square Residual (SRMR). It has been suggested that values greater than or equal to 0.90 for CFI and TLI and values lower than or equal to 0.08 for RMSEA and SRMR indicate a good fit [48].

We followed the two-wave model suggested by Maxwell and Cole to examine the hypothesized mediated processes [49]. Following the three-step procedure using the two-wave study design recommended by Cole and Maxwell [50], the partial mediation effect can be identified by testing the significance of αβ estimate (i.e., path α: the product of unstandardized cross-lagged path coefficients of teacher-student relationships at T1 on need satisfaction at T2; path β: the product of unstandardized cross-lagged path coefficients of need satisfaction at T1 on prosocial behavior at T2).

## 3. Results

The means, standard deviations, and bivariate zero-order correlations for all study variables at T1 and T2 are presented in Table 1. As expected, teacher-student relationships, need satisfaction, and prosocial behavior significantly correlated each other at T1 and T2.

### 3.1. Longitudinal Measurement Invariance Test

We used confirmatory factor analysis to examine measurement invariance across time. First, we estimate an unconstrained model (configural invariance) that included all latent factors at T1 and T2 and allowed their loadings, intercepts, and item variances to vary freely over time. Then, we compared this unconstrained model with a constrained model (weak factorial invariance) in which the corresponding factor loadings were allowed to be equal over time and no constraints on the intercepts. The results showed that the fit of constrained model (χ^2^/*df* (1306.40/667) = 1.96, CFI = 0.97, TLI = 0.96, RMSEA = 0.05, SRMR = 0.04) was not significantly worse than the unconstrained model (χ^2^/*df* (1285.52/653) = 1.97, CFI = 0.97, TLI = 0.96, RMSEA = 0.05, SRMR = 0.03; Δχ^2^/Δ*df* (20.88/14) = 1.49, *p* > 0.05), suggesting our measurements remained invariant over time.

### 3.2. Testing the Proposed Model

We test the reciprocal influences among teacher-student relationships, basic need satisfaction, and prosocial behavior using latent cross-lagged-panel analysis. We sequentially examined the estimation of four models: First, the baseline stability model (M0), where autoregressive paths are only included, demonstrated a good model fit for the data, with all autoregressive paths between T1 and T2 variables being significant, suggesting the long-term stability of constructs across time. Second, we examine the causality model (M1) in which the cross-lagged paths from T1 predictors (closeness and conflict) to T2 mediators (basic need satisfaction) and T1 mediators to T2 outcome (prosocial behavior) were added while controlling for the autoregressive effects over time. The results showed that M1 provided a satisfactory fit to the data and a significantly better fit than M0. Third, we calculated the reversed causation model (M2), which included cross-lagged paths from T1 mediators to T2 predictors and T1 outcome to T2 predictors and T2 mediators. This model also fits significantly better to the data than M0 (Δχ^2^/Δ*df* (72.74/11) = 6.61, *p* < 0.001). Finally, the reciprocal model (M3), which includes the combination of the causality model (M1) and reversed causation model (M2), showed an excellent fit to the data. As illustrated in Table 2, the model comparisons showed that the reciprocal model (M3) demonstrated a better fit to the data than the stability model (M0: Δχ^2^/Δ*df* (207.85/22) = 9.45, *p* < 0.001), causality model (M1: Δχ^2^/Δ*df* (43.39/11) = 3.94, *p* < 0.001), and reversed causation model (M2: Δχ^2^/Δ*df* (135.11/11) = 12.28, *p* < 0.001), suggesting that reciprocal model (M3) was the best fitting model.

The reciprocal model (M3), as shown in Figure 1, presents all statistically significant paths. When controlling for the autoregressive effects, T1 closeness positively predicted T2 prosocial behavior, and T1 prosocial behavior positively predicted T2 closeness, indicating a reciprocal relation between close teacher-student relationship and adolescents’ prosocial behavior. Contrary to our prediction, the reciprocal relation between conflict and prosocial behavior is not significant. These results supported H1 and not H2. Regarding Hypothesis 3, T1 competence and relatedness need satisfaction was positively related to T2 prosocial behavior, but autonomy need satisfaction was not significantly related to T2 prosocial behavior.

Based on the significant paths, we tested the partial mediating effects of relatedness need satisfaction in the causal effect of teacher-student relationships on prosocial behavior using bootstrapping analyses (5000 samples; 95% bias-corrected confidence interval). If the confidence interval excludes zero, the mediating effects are assumed to be statistically significant. Two indirect paths were significant. (1) T1 closeness → relatedness need satisfaction → T2 prosocial behavior (estimate = 0.04, SE = 0.02, *p* < 0.05, 95% IC: 0.02–0.08); (2) T1 conflict → relatedness need satisfaction → T2 prosocial behavior (estimate = −0.05, SE = 0.02, *p* < 0.01, 95% IC: −0.11–−0.02). Thus, Hypothesis 4c and 5c were supported.

## 4. Discussion

This study aimed to explore the reciprocal effects between teacher-student relationships, need satisfaction, and prosocial behavior. We applied the relationships motivation theory (RMT) within SDT to explain how teacher-student relationships predicted adolescents’ prosocial behavior over time by influencing students’ basic psychological need satisfaction (autonomy, competence, and relatedness satisfaction). RMT posits that the quality of interaction with important figures has a reciprocal relation with individuals’ satisfaction of basic psychological needs over time [8], which, in turn, are related to adolescents’ social development. However, fewer studies have empirically validated this assumption. Thus, based on a full cross-lagged model, our results indicated that: (1) closeness has a reciprocal effect on adolescents’ prosocial behavior, and conflict has no significant effect on prosocial behavior; (2) closeness positively predicted competence and relatedness need satisfaction but not autonomy need satisfaction, whereas conflict negatively predicted all three psychological need satisfaction. Moreover, the bidirectional effects between competence and relatedness need satisfaction, and prosocial behavior were significant; (3) relatedness need satisfaction partially mediated the associations between both closeness and conflict and adolescents’ prosocial behavior. To our knowledge, this is the first study to examine the temporal relationships among these variables during adolescence, and the underlying mediating mechanism of basic need satisfaction in these relationships.

### 4.1. The Reciprocal Associations between Teacher-Student Relationships and Adolescents’ Prosocial Behavior

As expected, we found a reciprocal association between close teacher-student relationship and prosocial behavior. Specifically, students who reported higher levels of a close relationship with their teacher are likely to engage in more prosocial behavior, which, in turn, contributes to warm and close teacher-student relationships over time. These results are aligned with previous studies that found a positive effect of closeness on prosocial behavior as well as the predicting role of prosocial behavior in enhancing the close relationship with teachers [6,11]. These results can be attributed to the fact that students are commonly promoted and expected by the teacher to display more prosocial behavior in the school. Further, students tend to engage in prosocial behavior in order to obtain and sustain a close and intimate relationship with important others, such as teachers, satisfying their psychological need for relatedness. However, in contrast to our prediction, conflict failed to show a significant effect on prosocial behavior over time, which slightly contrasts with previous longitudinal research that indicated a causal effect of teacher-student relationship conflict on prosocial behavior in young children over time. Nevertheless, research on early adolescence has obtained the same result as this study [26]. This result may be explained by the reason for adolescents engaging in prosocial behavior. The reason adolescent display more prosocial behavior is to establish and maintain a positive relationship with important figures (e.g., teachers and peers), but importantly this is the means by which they gained high level of social status in classroom that is commonly valued by middle adolescent students [51]. Thus, positive and close teacher-student relationship can promote adolescents engaging more prosocial behavior, however, conflictual teacher-student relationship does not necessarily result in decreased prosocial behavior. Indeed, the current findings highlight the facilitating role of close teacher-student relationship in enhancing adolescents’ prosocial behavior, which in turn positively predicted the establishment and maintenance of warm and close relationships with their teachers.

### 4.2. Teacher-Student Relationship and Basic Need Satisfaction

Consistent with the proposition posed by RMT in which a high-quality close relationship leads to the satisfaction of basic need for relatedness, our results indicated the causal relationship between teacher-student relationship and basic psychological need satisfaction. Specifically, close teacher-student relationship positively predicted adolescents’ subsequent perception of need satisfaction for competence and relatedness over time. These results are in line with previous cross-sectional and longitudinal research [31,34,52], suggesting that having a positive and high-quality relationship with important figures can satisfy adolescents’ basic psychological needs, especially relatedness. Furthermore, in contrast with the prior study reporting a significant reciprocal relationship between close teacher-student relationship and autonomy need satisfaction [31], we found positive reciprocal effect between competence need satisfaction and close teacher-student relationship. This finding suggests that positive and high-quality relationship is likely to provide students with more positive feedback about their ability and enhance their belief in controlling and interacting with the external environment. Moreover, this high level of competence need satisfaction will increase intrinsic motivation and social competence [53], which in turn promoted them to initiate and establish positive relationships with their teachers. Nevertheless, we also found that conflict negatively predicted adolescents’ satisfaction of all three psychological needs over eight months, but not the other way around. This finding extends RMT to suggest that negative aspects of teacher-student relationship such as conflict serves as a detrimental factor undermining the satisfaction of all three psychological needs and resulting in a lack of need satisfaction.

### 4.3. The Mediating Role of Basic Need Satisfaction

This study revealed that relatedness need satisfaction mediated the link between teacher-student relationships and later prosocial behavior. However, we did not find the mediating role of autonomy and competence need satisfaction in these links, although the direct effects of competence and relatedness need satisfaction on prosocial behavior were significant. Specifically, close teacher-student relationship positively predicted satisfaction of the need for relatedness, which, in turn, promotes adolescents’ engagement in more prosocial behavior. In contrast, teacher-student relationship conflict negatively predicted relatedness need satisfaction, decreasing the likelihood of prosocial behavior over time. These results were in line with existing research showing that relatedness need satisfaction served as a mediator in the development of prosocial behavior [38,40,54]. Our findings suggested that relatedness need satisfaction was more salient for fueling prosocial behavior than other psychological needs because it contributed to a higher level of empathy and social competence that promotes adolescents’ prosocial behavior [55]. Conversely, a lack of need satisfaction caused by teacher-student relationship conflict tends to result in decreased moral responsibility and amotivation [56,57], further undermining adolescents’ willingness to display prosocial behavior. Moreover, these findings have supported the SDT view, which proposes that basic psychological need satisfaction act as a motivational mechanism through which social environment factors related to an individual’s well-being and social behavior [29]. Overall, these results extended prior work by offering empirical evidence for the mediating role of basic psychological needs in explaining why positive and negative manifestations of teacher-student relationships were related to adolescents’ prosocial behavior over time.

## 5. Limitations and Future Research

In this study, data were collected from a relatively homogeneous sample of adolescents from a single secondary school in China using the convenience sampling technique, which may raise issues about the generalizability of the findings to other age cohorts. Future research would benefit from replicating and extending this study in a larger sample and different research contexts.

Although this study adopted the cross-lagged study design, which to some extent avoids the shortcomings raised by cross-sectional design, it still precludes drawing causal inferences between variables. Therefore, our results should be interpreted with caution. We encourage future research to replicate and examine our results using three or more-wave longitudinal or experimental study designs to make more robust causal inferences.

We relied on student reports to assess the quality of teacher-student relationships, which may exist single-source bias or over-flatted effect. As such, a meta-analysis study indicates that peer and teacher reports of dyadic teacher-student relationships could provide new insights into the role of teachers as important socialization figures [7]. Therefore, we recommend that future research include self- and peer-reported measures of studied constructs, which informants can provide a reliable assessment of the teacher-student relationship.

Although our results support the mediating role of basic psychological need satisfaction, extant research suggests that need frustration and need satisfaction represent two distinct motivational constructs instead of two sides of the same coin [28,58]. Thus, simultaneously examining the mediating effects of both psychological needs may offer new insight into the motivational mechanism of basic psychological needs in prediction of prosocial behavior.

## 6. Practical Implications

The present findings offer important practical implications for school intervention practice exploring ways to foster adolescents’ prosocial behavior.

First, this study highlights the importance of building and promoting teacher-student relationship closeness and reducing conflict. Our findings indicated a positive feedback loop between close teacher-student relationship and prosocial behavior over time. Specifically, establishing and maintaining a close and supportive relationship with teachers is conducive to promoting adolescents’ prosocial behavior that, in turn, fosters a close teacher-student relationship. Thus, schools should implement evidence-based intervention practices to improve the quality of teachers’ interactions with students. For example, research on teacher-student relationship invention has supported the effectiveness of Establish-Maintain-Restore (EMR) [59]. Researchers adopted the Classroom Check-Up programmer to increase teachers’ supportive comments and decrease conflict and reprimanding remarks [60].

Second, our findings suggest that in order to foster adolescents’ prosocial behavior, school and teachers should attach high importance to facilitating the satisfaction of adolescents’ psychological need for relatedness by creating a positive classroom interpersonal context. This study lends support to the mediating role of relatedness need satisfaction in transmitting the beneficial effect of closeness and the detrimental effect of conflict on prosocial behavior. Therefore, we recommend that schools provide multiple classes, such as social and emotional learning courses, enabling teachers to grasp more social skills in ways that fulfill adolescents’ basic psychological needs [61].

## 7. Conclusions

This study adopted RMT to investigate whether exists a reciprocal effect between teacher-student relationships and adolescents’ prosocial behavior and whether basic need satisfaction mediates the association between teacher-student relationships and prosocial behavior. In particular, we found positive reciprocal associations between closeness and adolescents’ prosocial behavior over time. However, the reciprocal association between conflict and adolescents’ prosocial behavior was not significant. Moreover, relatedness need satisfaction was found to only partially mediate the associations between closeness and conflict and adolescents’ prosocial behavior. These findings highlight the facilitating role of close relationship with teacher and the hindering role of conflict relationship with teacher in affecting relatedness need satisfaction, which in turn influence adolescents’ prosocial behavior.

## Figures and Tables

**Figure 1 ijerph-19-14840-f001:**
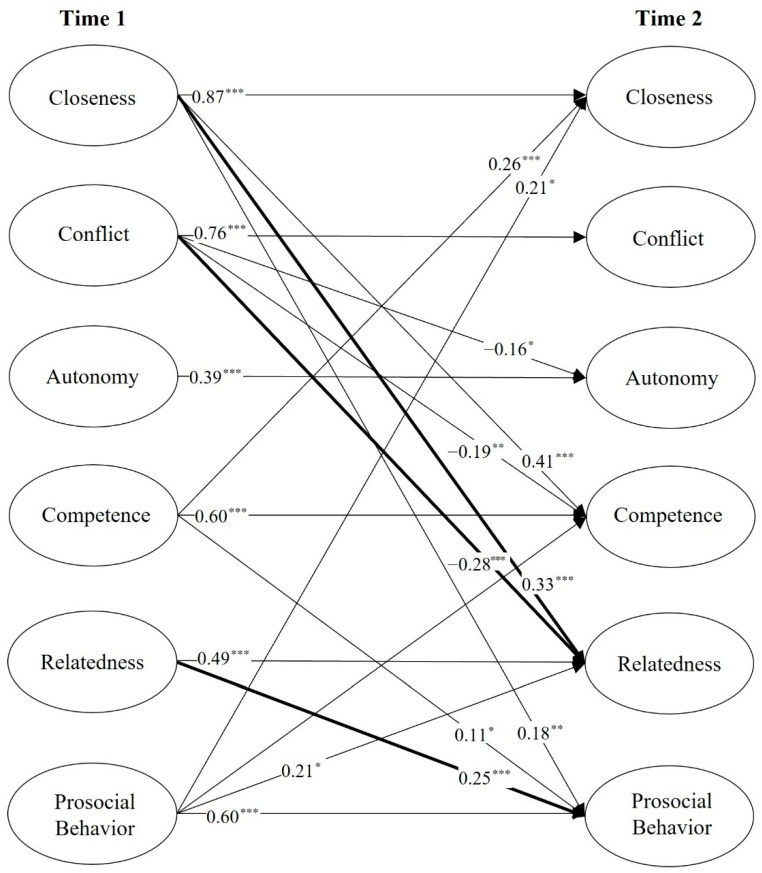
The final model. Note: Autonomy = autonomy need satisfaction; Competence = competence need satisfaction; Relatedness = relatedness need satisfaction; all the reported coefficients are standardized. Solid lines indicate significant paths. ** p* < 0.05, *** p* < 0.01, **** p* < 0.001.

**Table 1 ijerph-19-14840-t001:** Means, Standard Deviations, and Bivariate Correlations at Time 1 (T1) and Time 2 (T2).

Variables	*M*	*SD*	1	2	3	4	5	6	7	8	9	10	11
1.T1 Closeness	3.39	0.89	1										
2.T1 Conflict	1.86	0.63	−0.56	1									
3.T1 AS	2.27	0.47	0.26	−0.14	1								
4.T1 CS	3.45	0.81	0.43	−0.29	0.54	1							
5.T1 RS	3.92	0.74	0.64	−0.43	0.53	0.69	1						
6.T1 Prosocial	0.18	0.16	0.51	−0.31	0.33	0.59	0.63	1					
7.T2 Closeness	3.28	0.84	0.71	−0.50	0.36	0.51	0.65	0.45	1				
8.T2 Conflict	1.90	0.46	−0.53	0.71	−0.21	−0.32	−0.44	−0.29	−0.61	1			
9.T2 AS	2.45	0.47	0.24	−0.22	0.35	0.24	0.31	0.15	0.29	−0.28	1		
10.T2 CS	3.25	0.83	0.58	−0.48	0.36	0.47	0.61	0.42	0.61	−0.43	0.39	1	
11.T2 RS	3.58	0.65	0.64	−0.57	0.29	0.43	0.59	0.38	0.64	−0.51	0.37	0.75	1
12.T2 Prosocial	0.26	0.83	0.59	−0.34	0.42	0.65	0.73	0.78	0.55	−0.32	0.19	0.49	0.47

Note: All correlations are significant at *p* < 0.001. AS = autonomy need satisfaction; CS = competence need satisfaction; RS = relatedness need satisfaction; Prosocial = prosocial behavior.

**Table 2 ijerph-19-14840-t002:** Summary of model fit indices.

Models	χ^2^	*df*	RMSEA [95%CI]	CFI	TLI	SRMR	Δχ^2^/Δ*df*
M0: Stability model	1248.13	697	0.042 [0.039, 0.046]	0.962	0.958	0.069	
M1: Causality model	1083.67	686	0.036 [0.032, 0.040]	0.973	0.969	0.040	M1-M0: 14.95 ***
M2: Reversed causation model	1175.39	686	0.040 [0.036, 0.044]	0.967	0.962	0.046	M2-M0: 6.61 ***
M3: Reciprocal model	1040.28	675	0.035 [0.031, 0.039]	0.975	0.971	0.034	M3-M0: 9.45 ***
							M3-M1: 3.94 ***
							M3-M2: 12.28 ***

Note: **** p* < 0.001.

## Data Availability

Some or all data and models that support the findings of this study are available from the corresponding author upon reasonable request.

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
