# Peer review of "Longitudinal Associations between Teacher-Student Relationships and Prosocial Behavior in Adolescence: The Mediating Role of Basic Need Satisfaction"

_ijerph, 2022, doi:10.3390/ijerph192214840_

Round 1

Reviewer 1 Report

  • This is an interesting paper that investigates whether the associations between teacher-student relationships and prosocial behavior are bidirectional and if psychological needs satisfaction mediates this relationship among adolescents.
  • General concept comments:
  • The background covers the literature, data are analyzed in a sound way, the results make sense, and the discussion is interesting. Still, I have a few concerns.

  • Specific comments 
  • When relationships motivation theory (RMT) is mentioned the first time on p. 1, a reference is missing.
  • When teacher-student relationships are introduced on p. 2, a thorough background is missing. For example, many papers refer to the two dimensions Influence (Dominance—Submission) and Proximity (Opposition—Cooperation), others refer to Directive, Authoritative, Tolerant/Authoritative, Tolerant, Uncertain/Tolerant, Uncertain/Aggressive, Drudging and Repressive, and when closeness and conflicts are referred to, a third dimension called negative expectations is often discussed. The authors need to include this theoretical background.
  • In the method, more information on the context is needed. How big were the schools, what did the students study, how big are the classes on average, what socio economic background do students have etc.
  • Why was the follow-up carried out 8 months after the first wave? An argument for this would be helpful.
  • The decision to create parcels for the measures should be explained more. What are the limitations and advantages in doing this?
  • The discussion is overall very good, but I would like to see more of a theoretical discussion. How do the results relate to existing theories, e.g., how can it better explain the theoretical bases of RMT? As it is now, the discussion is mostly empirically driven.
  • Limitations and practical implications in general good, but I think there is a need for a broader discussion here. For example, do the results come from a typical school? or are there specifics here that need to be considered? Can the results be generalized to other cultures and countries? And when the authors discuss how the results could be applied in a school setting, EMR is given as an example. However, that model is based on middle school, why the current study focused on adolescents, and this is not acknowledged in this discussion as I could see.
  • I think the authors need to explain what is meant with "classes, such as social and emotional learning classes", as I have not heard about such classes before. Perhaps it is common in China, but it is not common in Europe.
  • The sentence on l. 124, p. 2, makes no sense ("it is insufficient 124 to ensure high-quality relational bonds with others only if all three basic psychological 125 needs are satisfied simultaneously [28]").
  • I think it is easier to read this paper if it says, "close teacher-student relationships” rather than "teacher-student relationship closeness".

  • General comments
  • The manuscript is pretty clear, relevant for the field and presented in a well-structured manner. However, there are problems with the handling of English many times. In the results, present and past tense are not used correctly, and it would have been preferable if the results had been presented in past tense throughout, e.g., we estimated (not we estimate), we tested (not we test) etc.
  • Finally, I wonder why Informed Consent Statement is not appliable to this study.

Author Response

Please see the attachment “Response to reviewers1”

Reviewer 2 Report

This work set out to investigate a model for the reciprocal relationships between the quality of teacher-student relationship and prosocial behavior in adolescents, and if those relationships are mediated by the satisfaction of basic motivational needs. They collected a relevant sample (though limited to one school) and the statistical analyses are appropriate and well-explained (though the interpretation of some results are dubious); see my comments below. Other than that, there are some aspects of the introduction and discussion that need to be made clearer; in addition to my comments below, please consider revising the use of the English language throughout the manuscript.

Introduction

-        Why/ what different studying this subject in secondary education (Vs childhood and early adolescence)?

-        Please specify the role that will be given to basic psychological need satisfaction.

-        The sentence “This perspective has gained much empirical support from research among children and adolescents” should be based on previous empirical research (more so than the example that is given in sequence to that sentence).

-        Conflict in student-teacher relationship is defined twice (lines 51-52 and lines 65-66). It would be better if information was organized in such a way as to not be repeated.

-        It would be useful if the authors clearly distinguish what are their assumptions based on empirical research from what was found in previous works. For example, the sentence at lines 59-61 seems to refer to the authors proposal based on previous findings, but that is not clear.

-        It would also be useful if the authors clearly distinguish what studies are based on cross-sectional correlational (i.e., bidirectional association within the same time point) from cross-sectional regression analyses, and from longitudinal works. It is implicit that, for example, section 1.1 refers only to works focusing on cross-sectional regression analyses, but that is not clear.

-        The sentence at lines 75-77 may be an overstatement if previous findings refer to correlation or regression analyses.

-        Lines 90-91: aren’t these findings counter intuitive? Prosocial behavior associating with teacher-student relationship conflict? Why should this findings be present in childhood but not later in development? Explanations based on developmental stages seem relevant, because previous findings are inconsistent, and the authors propose to study a developmental stage that has been more scarcely considered.

-        Lines 110-114 and lines 124-126 are not clear. Please revise.

-        Lines 155-157: the authors present with the topic of the role of diver basic need satisfaction, which has not been introduced before. If this difference is to be meaningful to this work, it should be clearly put forth when reviewing previous literature – what was found in relation to each one of these basic needs.

-        Lines 172-173: Do studies 37 and 38 not refer to this topic?

-        The authors should appropriate discuss the limitation of having the independent variable and the mediating variable measured at the same time. Why doing the analyses like this and not considering the mediating variables at T2?

Methods

-        Please be more explicit about sample constitution (e.g., n and percentage for boys and girls, age for the complete sample). It might be interesting to know if respondents to T1 and T2 differ in any meaningful way from respondents to T1 only…

-        Were legal guardians of underaged students asked for consent for students’ participation in the work?

-        Participants were aged 12 to 15 years old. It this not considered early adolescence?

-        The measure of prosocial behavior is not clear. How was it measured as to make up a roughly continuous variable? How was it associated with the self-report questionnaires for each respondent?

-        Authors should present information on previous psychometric properties found for the instruments they are using. This will make it clear that they choose the best way to access their intended variables.

-        After reading the statistical analyses section, I am not sure the mediating variables were considered (only) at T1…?

Results

-        Lines 294-303: The measurement models for all variables were tested simultaneously?

-        Please note that the p value associated with the chi-square diff test is presented as p < .001, meaning that the fit of the constraint model is significantly worse than the fit of the unconstraint model, unlike what is stated in the text. Please revise.

-        After reviewing the chi-square diff test, please be explicit about the measurement models remaining invariant over time concerning weak/ metric invariance only.

-        In figure 1, all paths are significant, but some lines are bold and other are not. Why is that?

-        When comparing the models, again the p value presented is < .001… This usually means worsening of the model, which is not the case according to information presented in Table 2. Please explain/ revise.

Discussion

-        Lines 349-352 refer only to one direction of relationships – so it simplifies and does not accurately refer to the current work.

-        The mediating significant pathways referred to only one direction. That should be put in a more straightforward way, and not as part of bidirectional effects.

-        Lines 382-387 refer to why students may look for peers for positive relationships, but current findings indicate the relevance of positive relationships with teachers. So, the (absent) role of conflict with teachers is still not/poorly explained.

-         Lines 394 – this was not an experimental study, so referring to causal relationships doesn’t seem appropriate.

-        Lines 401-408: if the reciprocal effect was not found, then eh last part of this segment was not validated based on current findings. Please revise.

-        Lines 412: why are conflictual relationships more predictive than positive ones?

Author Response

Please see the attachment “Response to reviewers2”
